# Qualitative process evaluation from a complex systems perspective: A systematic review and framework for public health evaluators

**Elizabeth McGill**[1]*, **Dalya Marks**[2], **Vanessa Er**[1], **Tarra Penney**[3¤], **Mark Petticrew**[2], **Matt Egan**[2]

**1** Department of Health Services Research and Policy, London School of Hygiene & Tropical Medicine, London, United Kingdom, **2** Department of Public Health, Environments and Society, London School of Hygiene & Tropical Medicine, London, United Kingdom, **3** MRC Epidemiology Unit, Centre for Diet and Activity Research (CEDAR), University of Cambridge, Cambridge, United Kingdom

¤ Current address: School of Global Health, Faculty of Health, York University, Toronto, Canada
* elizabeth.mcgill@lshtm.ac.uk

**Data Availability Statement:** Data were extracted from the primary studies, all of which are published and are listed in Table 1.

## Abstract

### Background

Public health evaluation methods have been criticized for being overly reductionist and failing to generate suitable evidence for public health decision-making. A "complex systems approach" has been advocated to account for real world complexity. Qualitative methods may be well suited to understanding change in complex social environments, but guidance on applying a complex systems approach to inform qualitative research remains limited and underdeveloped. This systematic review aims to analyze published examples of process evaluations that utilize qualitative methods that involve a complex systems perspective and proposes a framework for qualitative complex system process evaluations.

### Methods and findings

We conducted a systematic search to identify complex system process evaluations that involve qualitative methods by searching electronic databases from January 1, 2014–September 30, 2019 (Scopus, MEDLINE, Web of Science), citation searching, and expert consultations. Process evaluations were included if they self-identified as taking a systems- or complexity-oriented approach, integrated qualitative methods, reported empirical findings, and evaluated public health interventions. Two reviewers independently assessed each study to identify concepts associated with the systems thinking and complexity science traditions. Twenty-one unique studies were identified evaluating a wide range of public health interventions in, for example, urban planning, sexual health, violence prevention, substance use, and community transformation. Evaluations were conducted in settings such as schools, workplaces, and neighborhoods in 13 different countries (9 high-income and 4 middle-income). All reported some utilization of complex systems concepts in the analysis of qualitative data. In 14 evaluations, the consideration of complex systems influenced

**Funding:** The study and its contributing authors (EM, DM, VE, TP, MP, and ME) were supported by the National Institute for Health Research (NIHR) School for Public Health Research (SPHR) under grant number: PD-SPH-2015. https://sphr.nihr.ac. uk. The funder had no role in study design, data collection and analysis, decision to publish, or preparation of the manuscript.

**Competing interests:** The authors have declared that no competing interests exist.

intervention design, evaluation planning, or fieldwork. The identified studies used systems concepts to depict and describe a system at one point in time. Only 4 evaluations explicitly utilized a range of complexity concepts to assess changes within the system resulting from, or co-occurring with, intervention implementation over time. Limitations to our approach are including only English-language papers, reliance on study authors reporting their utilization of complex systems concepts, and subjective judgment from the reviewers relating to which concepts featured in each study.

## Conclusion

This study found no consensus on what bringing a complex systems perspective to public health process evaluations with qualitative methods looks like in practice and that many studies of this nature describe static systems at a single time point. We suggest future studies use a 2-phase framework for qualitative process evaluations that seek to assess changes over time from a complex systems perspective. The first phase involves producing a description of the system and identifying hypotheses about how the system may change in response to the intervention. The second phase involves following the pathway of emergent findings in an adaptive evaluation approach.

### Author summary

#### Why was this study done?

- Process evaluations are used in public health to understand how and why an intervention works (or does not work), for which population groups, and in which settings.

- Process evaluations often use qualitative methods—such as interviewing people and observing people in their daily and work routines—in order to draw their conclusions.

- Researchers in public health have contended that we need to do research in a manner that considers the broader system in which policies and interventions take place—something we call a "complex systems perspective."

- To date and to our knowledge, there is no specific framework that describes how researchers can use a complex systems perspective when they conduct a process evaluation with qualitative methods.

#### What did the researchers do and find?

- We conducted a systematic literature review that looked for examples of qualitative process evaluations that self-identify as using a complex systems perspective to evaluate public health interventions.

- We found 21 different evaluations of many different types of public health interventions, including interventions to address student and employee health, sexual health, child development and safety, community empowerment, violence prevention, and substance use.

- We found that these evaluations describe the systems in which public health efforts take place but are less effective at analyzing how changes affecting health occur within these systems.

**What do these findings mean?**

- There is little evidence of a commonly shared understanding of how best to bring a complex systems perspective to process evaluations using qualitative methods, particularly, how to assess how interventions interact with a changing system.

- We developed a 2-phase framework to guide researchers who want to apply a complex systems perspective to qualitative process evaluations.

- This review excluded studies that do not self-identify as using a complex systems perspective so we may have missed literature that uses this perspective but not the associated terminology.

## Introduction

There has been a growing call [1] for the application of complex systems approaches to intervention planning, service delivery, and evaluation in order to aid understandings of intervention implementation and impacts in real-world environments [2–4]. Complex systems have been framed as a kind of antidote to reductionist approaches to health research [5]. Finding ways to bring a complex systems perspective to public health evaluation could, it is hoped, shed new light on how to address public health challenges in a complex world. A complex systems perspective can be applied to many different types of research design and methodology. In this paper, we focus on how such a perspective has been applied to process evaluations that utilize qualitative methods. The remainder of this section elaborates on what is meant by complex systems and process evaluations and discusses why qualitative methods are a particular area of interest for public health evaluators interested in complex systems.

### Complex systems

Systems are combinations of elements that interact. A distinction is often made between "complex" systems and systems that are "simple" or "complicated" [6–8]. What make complex systems unique are a number of attributes, including nonlinearity, their dynamic and unpredictable nature, and the ways in which they co-evolve with their environment and produce emergent outcomes [9–11]. Elements within a complex system (for example, individuals, organizations, activities, and environmental characteristics) interact with each other and are connected in nonlinear ways [6,12–14]. Over time, the behavior of system elements leads the individual elements and the system as a whole to adapt and co-evolve with the broader environment—that is, the system is dynamic [6,7, 12,13]. There may or may not be a central authority within the system, such as a president, local authority, or management team, but a complex system is assumed to adapt and behave in ways that cannot be reduced to simple, organizational hierarchies. Because of this, a complex system and its elements are considered to be self-organizing [6]. The individual interactions among system elements collectively

generate emergent, system-level behavior wherein the system displays attributes that cannot be reduced to its individual parts [2,6,12,15].

Research into complex systems takes place across academic disciplines and has roots in both *systems thinking* and *complexity science*. Although often grouped together because of some conceptual similarities, systems thinking and complexity science can be considered as distinct yet overlapping traditions [16,17]. Systems thinking may be best described as an orientation that prompts researchers to take a holistic, rather than reductionist view, of phenomena and study them in the context of their real-world systems that are open to and interact with surrounding systems. Systems thinking draws on theories, concepts, and methods from a range of disciplinary fields [18]. Complexity science, on the other hand, is more strongly rooted in the mathematical sciences and has drawn on complexity theory, which emphasizes uncertainty and nonlinearity, to create and refine specific methodological approaches to modeling complex systems in order to estimate and predict their emergent behavior over time. Systems thinking prompts researchers and practitioners to consider the boundaries of the system they are studying or in which they are working [19] and places an emphasis on the interactions and relationships between system elements and the system with its broader environment [1,6]. Further applying concepts from complexity science prompts a consideration of how those interactions create nonlinear chains of cause and effect, are unpredictable, unfold over time, and give rise to system-level emergent outcomes [20].

Complexity has been part of the vocabulary of public health evaluators for decades [16,21]. However, public health evaluations have tended to focus on the complexity of interventions rather than of the systems within which interventions are implemented [22]. A "complex intervention" is one that has a number of interacting parts, targets different organizational levels or groups of people, and aims to affect a number of outcomes [16,17]. In contrast, a complex systems perspective considers complexity as an attribute of the system. The intervention itself may also be complex, for example, a coordinated program of interventions that affect different parts of a system. However, simple interventions can also be theorized to have complex consequences if they are implemented within and interact with a complex system. For example, a single change in a law affecting the price of products that affect health (such as an alcohol or sugar sweetened beverage tax) can be described as an (initially) simple intervention that quickly becomes connected to a complex chain of interactions between industry, retailers, public opinion, consumer behavior, media and policy—each of which may have an impact on future implementation and effects of the intervention itself [15,23]. The way a complex system responds to an intervention may lead to emergent consequences that could amplify or dampen the intervention's impacts, change the characteristics and behavior of the system over time, and affect future decision-making [15,24]. From a complex systems perspective, the role of the evaluator is to make sense of the interplay between the complex system and the (simple or complex) intervention to help explain health and other impacts and inform future decisions about implementation [1].

## Process evaluations and qualitative methods

Traditional evaluations of simple or complex public health interventions often focus on measuring impacts on a single (or small number) of prespecified health and health-related outcomes [10]. However, impact evaluations alone offer little opportunity to explore the mechanisms behind an intervention's success or failure, particularly when impacts are unevenly distributed among different population groups. For this reason, other forms of evaluation, particularly process evaluation, have been developed and utilized in order to understand intervention implementation and the mechanisms by which interventions may lead to impacts

across a population [17,25]. There is no single definition of a process evaluation, but the Medical Research Council's (MRC) Guidance on Process Evaluations of Complex Interventions argues they "can be used to assess fidelity and quality of implementation, clarify causal mechanisms, and identify contextual factors associated with variation in outcomes" [26 p. 30]. A process evaluation is often, although not always, conducted alongside an outcome or impact evaluation that quantifies the impact of an intervention on a range of outcomes [16].

Process evaluations of public health interventions may benefit from an explicit adoption of a complex systems perspective. The application of systems thinking and insights from the complexity sciences can provide a means through which to evaluate and understand the non-linear ways in which interventions may lead to a number of impacts within a system. This could include impacts considered to be of interest when the evaluation is initially planned and impacts that emerge as potentially important as the evaluation progresses. By bringing an explicitly relational focus to the evaluation design and placing the wider context in the foreground of the analysis [24], a complex system approach to a process evaluation may help to make sense of intervention mechanisms within a real-world context. An explicit complex systems perspective may also help evaluators construct a narrative that explores the trajectory of a given system. This could include considering how the intervention acts as an event that prompts a series of changes in the way a complex system behaves [15]. Furthermore, it could include consideration of how the intervention itself changes, as system elements and the system as a whole adapt and respond to it [15,24].

Although process evaluations can include quantitative assessments of intervention outputs, they typically draw on a range of qualitative methods. Qualitative methods are well suited for unpacking complex causal chains, understanding changes in implementation, representing varying experiences of the intervention, and generating new theories to inform future decision-making [17]. Proponents of explicitly using complexity theory within qualitative designs argue doing so "has potential to capture and understand complex dynamics that might otherwise be unexplored" [27 p. 3]. Bringing a complex systems perspective to a qualitative process evaluation could have a range of methodological implications. For example, it could involve mapping the system of interest, a sampling strategy that seeks to recruit participants relevant to different parts of that system, a form of data collection geared towards assessing relationships within a system, and an analysis framework that incorporates concepts drawn from systems thinking and complexity science.

There is a large body of literature on quantitative methods for complex systems approaches and some examples of such methods being applied to the study of policies and interventions that may affect population health [28–33]. Many of these approaches build simulation models that estimate and predict the impact of interventions on outcomes of interest [34]. These approaches have been developed within the complexity sciences and include methods such as system dynamics modeling, microsimulation modeling, and agent-based modeling [3,20,35,36]. Although these methods may begin with some qualitative work, such as participatory workshops to map a system of interest, their aim is to generate quantitative estimates of future or hypothetical impacts [31]. Compared with quantitative methods, there is little consensus, and less has been written on how to explicitly draw on a complex systems approach for process evaluations that use qualitative methods. This represents an underdeveloped area for complex systems evaluation.

This systematic review therefore aimed to identify the concepts and methods currently used in public health evaluations that apply a complex systems perspective to process evaluations involving qualitative methods. Specifically, this review sought to answer 3 research questions: (1) What types of public health interventions have been subjected to process evaluations that use qualitative methods and apply a complex systems perspective? (2) What are the qualitative

methods used in this body of literature? (3) What concepts and theories associated with complex systems are used in process evaluations that use qualitative methods? Drawing on this body of literature, we then had a secondary aim of developing a framework for qualitative process evaluation from a complex systems perspective. We sought to develop an evaluative framework that researchers (working in academic or practice settings) can use as an overarching structure to guide evaluative efforts [37]. In our Discussion section, we therefore present our framework and provide some guidance for researchers on the potential role of qualitative data in identifying and understanding aspects of complexity within process evaluations.

## Methods

### Data sources and screening

Relevant process evaluations were identified through several different search methods. First, we conducted an expert consultation whereby we contacted 32 academics with an interest or experience in complex systems thinking and its application to public health and asked them to identify any relevant examples of complex systems evaluations. The academics were identified through an ongoing familiarization with the literature on complex systems and public health, as well as through our own professional networks. In the original consultation, we did not request permission to be named, but those who did provide permission during the review process are named in the Acknowledgments. We then identified 2 relevant systematic reviews on systems thinking and public health [35] and complexity theory applied to evaluation [20]. From the studies identified in these reviews, we selected evaluations that met our inclusion criteria (next). Finally, we conducted an electronic search covering January 1, 2014–September 30, 2019 using 3 databases: Scopus, Medline, and Web of Science. The search dates were set to capture evaluations published after the 2 systematic reviews. The electronic search strategy included terms and synonyms for systems thinking, complexity science, evaluation, and public health and was restricted to English-language publications. An example of the full search strategy can be found in S1 Text. This study is reported as per the Preferred Reporting Items for Systematic Reviews and Meta-Analyses (PRISMA) guideline (S1 PRISMA Checklist).

Titles and abstracts were screened initially by one reviewer, and all potentially relevant studies were independently screened by 2 reviewers. In cases in which a decision was not clear cut, or the reviewers disagreed, a discussion was held with a third reviewer. The review had 4 inclusion criteria, which we describe in more detail next. In brief, studies were included in the review if they (1) self-identified as taking a systems- or complexity-informed approach; (2) were relevant to public health; (3) were process evaluations of interventions with empirical findings; and (4) utilized qualitative methods.

Studies were eligible for inclusion if they self-identified as using a systems and/or complexity perspective at any stage of the evaluative process, including during the design, data collection, analysis, or interpretation phases. We took a broad view of public health to include upstream determinants of population health, which include alcohol, the built environment, community health, community safety, education, employment, environmental health, food, health promotion, housing, illicit substances, obesity, policing, regeneration, sexual health, social welfare, tobacco, trading standards, transport, and urban planning. Studies that covered topics not included in the aforementioned list were considered if they concerned population health; decisions in these instances were made between 3 reviewers. Studies concerning treatment in health service settings were excluded. Studies were only included if they reported empirical findings of a process evaluation; protocols and discussion pieces describing evaluations without presenting results were excluded. Process evaluations alongside outcome evaluations were eligible for inclusion, although our analysis focused solely on the process evaluation

component. Finally, studies were eligible for inclusion if they used qualitative methods, which included interviews, group interviews or focus group discussions, (participant) observation, document review, free form responses on questionnaires, and participatory and visual methods, including for example, mapping workshops and photography. Evaluations employing mixed methods (wherein qualitative data were integrated into the assessment of the intervention alongside other methods) were included, as long as there was a substantive component that generated and analyzed qualitative data. To operationalize this criterion, we considered the ways in which the mixed methods research was designed, and we included studies that generated qualitative and quantitative data concurrently to evaluate an intervention (*triangulation design*); studies in which the researchers primarily utilized a qualitative design with some supporting quantitative output or outcome data (*embedded design*); studies in which the qualitative data were used to make sense of intervention outcomes (*explanatory design*); or studies in which qualitative research was used to generate hypotheses about the intervention that could be tested quantitatively (*exploratory design*) [26,38]. Studies utilizing these mixed method designs were eligible for inclusion even if the authors did not label the design or describe the rationale for the chosen approach. A substantive qualitative component referred to the authors both describing the qualitative methods, including data collection and analysis, as well as presenting qualitative data. Covidence software was used to help facilitate the screening process [39].

### Data extraction and synthesis

The analysis began with an in-depth reading of, and familiarization with, the included studies, with specific attention paid to the ways in which they drew on systems thinking and/or complexity science and the methods utilized to achieve their evaluative aims. Data were extracted on each study using a template designed for this review. Specifically, data on the study's research question, public health area, country, intervention, the application of complex systems thinking, the methods and analytical approach, and system map (if presented) were extracted (see Table 1). The "complex systems perspective and evaluation stage" column shows how systems thinking and/or complexity science featured in each evaluation and at which stage in the evaluation (i.e., design, data collection, analysis). The system map column reports the studies that included a map of the system and describes what the map detailed. If the evaluators published a logic model, it is noted in this column. Where studies gave rise to more than one publication, we considered them "linked" and extracted data from across the identified studies. The data extraction process was completed by one reviewer and double checked by a second.

Alongside the data extraction process, a list of concepts from systems thinking and complexity science was generated through an ongoing familiarization with these bodies of literature. A number of papers and books that are frequently referenced within the public health literature on complex systems were selected during this familiarization period [1,6,7,9,12,15,22,40], and from this, a master list of systems and complexity terms was generated. Our aim was that this list captured the key principles associated with each of the traditions and could be used by those wishing to gain a familiarization with systems thinking and complexity science. We found that not all authors describe the same concepts within these traditions and they often use different language. As a result, there was a subjective element to generating the list with the research team making choices about which concepts to feature and how to define them. In particular, although many authors describe "context" as a key systems thinking concept, and we initially also included it in our list, we ultimately chose to exclude it due to its substantial overlap with many other concepts. "Context" describes the factors in the

**Table 1. Characteristics of the included studies.**

| Study | Aim | Public health area | Country | Complex systems perspective and evaluation stage | Qualitative methods | System map |
|---|---|---|---|---|---|---|
| Alfandari 2017 [43], Alfandari 2019 [44] | To qualitatively evaluate the extent to which a national reform in Israeli child protection decision-making committees strengthened professional judgment through introducing a new standard tools package into practice. | Social work | Israel | Systems approach utilized as a conceptual framework to inform design and analysis | Observations, semi-structured interviews, and review of case records and reports. | None |
| Bartelink and colleagues 2018 [47], Bartelink and colleagues 2019 [46] | To explore the processes through which HPSF and the school context adapt to one another in order to generate and share knowledge and experiences on how to implement changes in the complex school system to integrate school health promotion. | School health | Netherlands | Systems concepts informed research questions, program theory, data collection methods and analysis | Interviews, observations, document review, and informal conversations. | Bespoke system diagram depicting the program theory |
| Burman and Aphane 2016 [48] | To use the Cynefin framework to situate emergent knowledge action spaces into appropriate decision-making domains, to inform subsequent phases of a bio-social HIV/AIDS risk reduction project. | School health, sexual health | South Africa | Cynefin framework used to guide the analysis and further intervention development | Group exercise and semi-structured group interviews. | Cynefin framework diagram |
| Crane and colleagues 2019 [51,52] | To describe and apply a pragmatic approach to evaluating the Get Healthy at Work initiative in New South Wales, Australia. | Workplace health | Australia | Systems thinking informed evaluation design, research questions and analysis | Focus groups, in-depth interviews, and observations. | Bespoke system diagram depicting program implementation levels and interaction points and program implementation cycle |
| Czaja and colleagues 2016 [53] | To use a systems engineering approach to identify the requirements for implementing community programs to prevent drug or HIV sex risk behaviors. | Sexual health, substance use | United States | Used systems engineering approach to develop research questions and inform analysis | In-depth interviews. | Bespoke system diagram of system elements and levels |
| Dickson-Gomez and colleagues 2018 [54] | To examine the implementation of a national HIV combination prevention strategy in El Salvador funded by the Global Fund to Fight AIDS, tuberculosis and malaria. | Sexual health | El Salvador | Used a "dynamic systems framework" to analyze data | In-depth interviews. | Bespoke system diagram with elements and linkages |
| Durie and Wyatt 2013 [42] | To evaluate a learning program designed to create transformational community change. | Community empowerment and transformation | United Kingdom (England) | Complexity theory informed intervention and evaluation design, including research questions, sampling strategy and analysis | Semi-structured interviews, nonparticipant observation, and community sessions. | None |
| Evans and colleagues 2015 [49] | To use a formative process evaluation to examine how a school-based intervention aimed at improving children and young people's social and emotional competencies moved through different phases of innovation within the complex school system. | School health | United Kingdom (Wales) | Diffusion of innovation theory applied as theoretical framework in data collection and analysis stages | Semi-structured interventions and observations. | None |
| Figuerio and colleagues 2016 [55] | To describe the development and proof of concept process of the critical event card analytical tool and to apply it to the development of leisure infrastructure in a poor urban environment. | Health equity policy Physical activity | Brazil | Drew on actor-network theory and applied the "critical event card" as an analytical tool to situate intervention within a complex system | Study seminar to create critical event timelines, interviews, and document review. | Bespoke timeline of critical events with interactions between components |

(*Continued*)

**Table 1.** (Continued)

| Study | Aim | Public health area | Country | Complex systems perspective and evaluation stage | Qualitative methods | System map |
|---|---|---|---|---|---|---|
| Fisher and colleagues 2014 [57] | To assess the extent to which an alliance of health and human service networks was able to promote effective action on the social determinants in an Australian urban region. | Urban planning | Australia | Complex systems perspective applied to data collection tools, analysis and interpretation of findings | Questionnaire, short interviews, and semi-structured interviews. | Bespoke system diagram showing interaction of factors across and within levels of the system |
| Haggard and colleagues 2015 [59] | To identify factors that either promote or hinder implementation of a multicomponent"Responsible Beverage Service" program in Swedish municipalities. | Substance use | Sweden | Systems thinking informed intervention; applied The Consolidated Framework for Implementation Research (with systemic components) to analysis | Semi-structured interviews. | None |
| Kearney and colleagues 2016 [65] | To evaluate how multiple system layers interact and influence each other within a gender-based violence prevention program in schools and explore how the evaluation further affected program implementation. | Violence prevention | Australia | Whole system approach informed intervention; applied conceptual approaches from systems science to guide data collection and analysis | Focus groups, interviews, and audit tool. | None |
| Knai and colleagues 2018 [63] | To use a systems approach to make sense of the evaluative findings on the UK's Responsibility Deal in order to explore why the initiative did not reach its objectives. | Public-private partnership for health | United Kingdom (England) | Systems approach applied to the integration and analysis of data from several independent, but linked evaluation strands | Literature review, interviews, organizational case studies, document review, media analysis, and analysis of pledges. | Causal-loop diagram Logic model |
| McGill and colleagues 2016 [60], Sumpter and colleagues 2016 [61] | To determine how a systems perspective can be used to explore the intervention's intended and unintended consequences within the local system and the effect of the intervention on alcohol availability. | Substance use | United Kingdom (England) | Systems perspective informed evaluation design and sampling strategy; complexity concepts used to generate research questions and structure analyses | Interviews, focus group, and local authority audits. | Bespoke system diagrams showing possible pathways to impact |
| Orton and colleagues 2017 [64] | To assess how a systems approach can be used to help understand how change processes that emerge as area-based empowerment initiatives embed and co-evolve within a series of local contexts. | Community empowerment and transformation | United Kingdom (England) | Systems approach used to inform sampling strategy and to inform analysis | Document review, interviews, observations, group exercises, focus groups, and participatory mapping. | None |
| Pérez-Escamilla and colleagues 2018 [62] | To examine the process of scaling up 3 major country-level early childhood development programs through the application of a "complex adaptive systems" framework. | Child development | Chile, India, South Africa | Used complex adaptive system constructs to develop data collection tool and used framework to guide the analysis | In-depth interviews and document review. | None |
| Rothwell and colleagues 2010 [41] | To assess the implementation of the WNHSS at national, local, and school levels, using a systems approach drawing on the Ottawa Charter. | School health | United Kingdom (Wales) | Intervention and setting conceptualized as complex adaptive system; socio-ecological model used to guide design, sampling strategy and analysis of findings | Document review, interviews, workshops, and observations. | Bespoke system diagram of the system structure |
| Schelbe and colleagues 2018 [45] | To describe the application of systems theory as a framework for examining a college campus-based support program for former foster youth. | Social work | United States | Applied systems theory to evaluation design and analysis and interpretation of findings | In-depth interviews and member checking. | None |

*(Continued)*

**Table 1.** (Continued)

| Study | Aim | Public health area | Country | Complex systems perspective and evaluation stage | Qualitative methods | System map |
|---|---|---|---|---|---|---|
| Shankardass and colleagues 2018 [56] | To present a systems framework to evaluate the implementation of Health in All Policies initiatives and to apply the framework to a case study of the Finnish policy "Health 2015." | Health equity policy Substance use | Finland | Applied a framework informed by systems thinking and realism to the analysis of data | Literature review and interviews. | Bespoke system diagram of the system structure |
| van Twist and colleagues 2015 [58] | To use a case of urban regeneration projects in the Netherlands to account for the "by-effects" of policy. | Urban planning | Netherlands | Developed framework informed by a complexity concept ("by-effects") which informed data collection methods and was used to structure analysis | Narrative interviews. | None |
| Walton 2016 [50] | To retrospectively explore the extent to which complexity concepts were applied in an evaluation of a school health promotion intervention. | School health | New Zealand | Applied complexity frame of reference to previous evaluation findings | Document review and key informant interviews. | None |

HPSF, Healthy Primary School of the Future; WHNSS, Welsh Network of Healthy School Schemes.

environment that affect the system, particularly historical, temporal, geographical, political, and social factors [13]. As a result, arguably the entire system represents the "context," and it therefore does not represent a meaningful category when trying to describe and analyze a changing system. In addition, we recognize that there is conceptual overlap between many of the concepts and that the boundaries between them may be somewhat fluid. In the Discussion section a glossary of terms and how they might be applied within a process evaluation using qualitative methods are presented.

## Critical appraisal

No tools exist to assess the quality of process evaluations informed by a complex systems perspective. Therefore, for this review, we critically appraised how systems thinking and complexity science were employed in each paper. Specifically, we assessed the degree to which each study identified through the search strategy described, captured, measured, or applied each concept in a meaningful way. The decisions were depicted using a traffic light color scheme. A green color code was applied when a study explicitly applied a concept at any stage of the evaluation process, including the design and planning stage, data collection, analysis, or interpretation. For example, a study would receive a green code if it explicitly described the boundaries of the system under inquiry at any stage in the evaluation. Evaluators might use the idea of boundaries, for instance, to shape the evaluation scope by designating clear system boundaries to bound the evaluation, or the concept might be applied within the interpretation of the data, to gain, for example, an understanding of how system elements view the boundaries of their own system. A yellow coding represented a study in which there was some attempt to apply a concept, but it was limited or addressed in an implicit manner. A red color code represented instances in which the concept was not utilized. The aim of this appraisal was not to be overly critical about individual studies but rather to understand the ways in which concepts from systems thinking and complexity science are applied in this body of literature. This process required us to make judgments, and in some instances, the decisions were not necessarily clear cut. In order to increase the validity of this process, 2 reviewers (EM and DM; or EM and ME) independently assessed each study, and disagreements were reconciled through discussion.

## Results

### Evaluation characteristics

A total of 21 unique evaluations (in 25 separate publications) were identified (see Fig 1). Their characteristics are presented in Table 1, and in-depth descriptions of 2 evaluations, one rooted in systems thinking [41] and another in complexity science [42], are presented in S2 Text. The in-depth descriptions were written to give clear examples of how these approaches have been

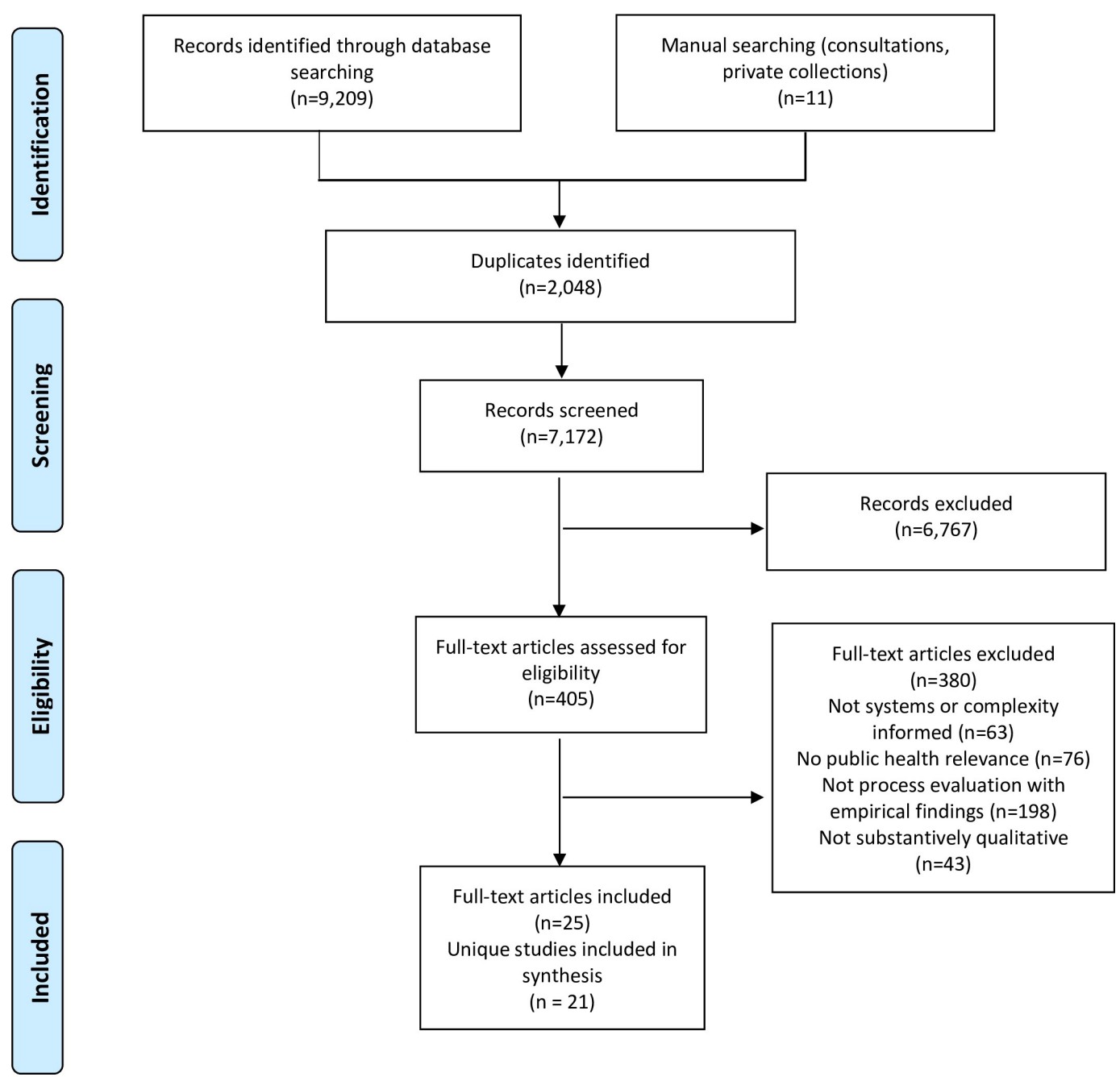

**Fig 1. Flow diagram for inclusion of studies.**

applied in practice. A range of public health topics were represented in the sample, including social work [43–45], school health [41,46–50], workplace health [51,52], sexual health [48,53,54], health equity policy [55,56], urban planning [57,58], substance use [53,56,59–61], child development [62], public–private partnerships [63], community empowerment and transformation [42,64], and violence prevention [65]. The studies were conducted in 13 countries, which included 9 high-income and 4 middle-income settings: Australia [51,52,57,65], Brazil [55], Chile [62], El Salvador [54], Finland [56], India [62], Israel [43,44], the Netherlands [46,47,58], New Zealand [50], South Africa [48,62], Sweden [59], the United Kingdom [42,49,60,61,63,64,41], and the United States [45,52].

The primary studies in this review were notable for their diversity in terms of the theories and frameworks used to inform the evaluation design and the focus of the analysis. Prominent theories included explicit applications of complexity theory [42,50,60] and diffusion of innovation theory [49]. Studies also used a number of frameworks to structure the analysis and to draw out evaluative findings. This included existing frameworks such as the Cynefin framework [48], Consolidated Framework for Implementation Research [59], a complex adaptive systems framework [54,62], and the socioecological model [41]. Other evaluations featured bespoke frameworks for analysis, including ones that focused on the role of critical events in an intervention's trajectory [55], a systems framework focusing on governmental subsystems [56], and a framework that was used to identify and categorize different types of "by-effects" or unintended consequences [58].

The process evaluations in this literature base varied in terms of the stage of evaluation planning and conduct in which they drew on complex systems thinking concepts and frameworks. Although the reporting was not always clear, 14 evaluation teams used some facets of systems thinking and complexity science when planning and designing their evaluations [41–47,49,51–53,57,58,60–62,64,65], which ranged from asking systems-oriented research questions to informing the sampling strategy (e.g., a conscious effort to sample different elements or from different levels within the system) and data collection tools (i.e., interview topic guides). Other evaluators used complex systems concepts, theories, or frameworks solely to structure their analyses [48,50,54–56,59,63].

The evaluations identified also drew on a wide range of qualitative methodologies. Ten studies applied a case study design [41–45,50–52,56,60–62,64]. The nature and boundary of a case varied from evaluation to evaluation. Some studies ($n$ = 3), for example, defined a case based on geographical boundaries, and each case represented a geographical locality [42,60,61,64]. Other case study examples included individual families [43,44] or schools [41] or the specific application of a policy [56].

Evaluators utilized a number of different methods for data collection, and 13 applied a mixed methods approach, which included using multiple qualitative data collection methods [41–45,48–50,55–58,62,64]. Seven studies employed a mix of qualitative and quantitative methods [46,47,51–53,59–61,63,65], although all of these studies had substantive qualitative findings. Not all evaluators articulated their rationales for choosing and combining certain qualitative methods, but in general, the different methods were employed to access, understand, and analyze different elements, structures, and relationships within the system. For example, speaking to a range of different actors within the system, through interviews (semi-structured, in-depth, or narrative) and focus groups [41–65], was used to assess different perspectives about an intervention, relationships, and theories of change within the broader system and to make sense of system trajectories. Documentary review and analysis were also relatively common, being used in 7 studies [41,43,44,46,47,50,62–64], and a range of documents were reviewed including media reports, community plans, evaluation documents, and case reports. Documents were used to understand intervention development and

implementation and to generate data at different levels within systems, for example, with some evaluators choosing to review national-level documentation and subsequently conduct regional or local-level interviews [41]. Seven of the evaluations identified also conducted both participant and nonparticipant observation, which ranged from observations of meetings to community events [41–44,46,47,49,51,52,64]. In addition to these researcher-led qualitative methods, some evaluators ($n$ = 10) utilized more participatory research techniques, including research seminars and workshops, mapping exercises, the creation of intervention timelines, and other types of group exercises [41,42,48,55,64]. Participatory methods were utilized both as a means of bringing in the perspective of those affected directly by the intervention, as well as a method to check and present interim findings.

Several of the identified process evaluations were conducted alongside or after impact/outcome evaluations of the same intervention. Knai and colleagues integrated data from several evaluative strands including impact and process evaluations [63]. Five studies reported accompanying outcome evaluations, but those results were not presented alongside the process evaluation reports [43,44, 46,47,59,64]. Three studies presented outcome data alongside their process evaluations [50–52, 60,61]. Finally, 2 papers reported independent outcome evaluations that were not linked to their own process evaluations [49,58].

The identified evaluations varied in the extent to which they produced and utilized system maps; 11 produced system maps of some description [41,46–48,51–57,60,63]; of these, only one used a formal system mapping technique: a causal-loop diagram [63]. The other system maps were bespoke maps that depicted different types of logic models [60,63], maps of the system structure [41,53,54], and maps that showed interactions between system elements [51,54,55,57].

## Application of concepts from systems thinking and complexity science

Evaluations varied in the extent to which they applied concepts from systems thinking and complexity science to their evaluation design or analysis and concepts from systems thinking were utilized to a far greater extent than complexity concepts. Fig 2 shows this using a traffic light coloring scheme. The figure is structured with different concepts from systems thinking and complexity science in each of the columns. The concepts are presented as belonging along a continuum, with systems thinking on the far left-hand side and complexity science on the far right-hand side. Moving along the spectrum, from systems thinking to complexity science, represents a movement from static to dynamic. Key systems thinking concepts, on the left-hand side of the figure, are the structure of a system, its elements, and the relationships between them. Utilizing these allows researchers to create relatively static depictions of a system. Moving toward the middle of the figure, concepts from complexity science are introduced, which include attributes and dimensions of an intervention, and then a system undergoing change. The far right-hand side of the figure includes concepts that feature within the complexity sciences to computationally model complex systems in order to simulate and predict behavior and outcomes and to understand an evolving system.

The evaluations identified in this review consistently applied key concepts from systems thinking: the identification and description of the system structure, including the different system elements and their differing perspectives. Thinking systemically also means making sense of the boundaries of a system and making decisions about what constitutes "the system" and what might be considered within or outside of the system. Although system maps are not a necessary element of systems thinking, they can be helpful for making sense of and depicting system boundaries, as articulated by both those acting within the system ("first-order" boundary judgments) and those studying it ("second-order" boundary judgments) [66]. Few evaluations ($n$ = 3) in the sample [42,45,64] had explicit discussions of boundaries and the ways in

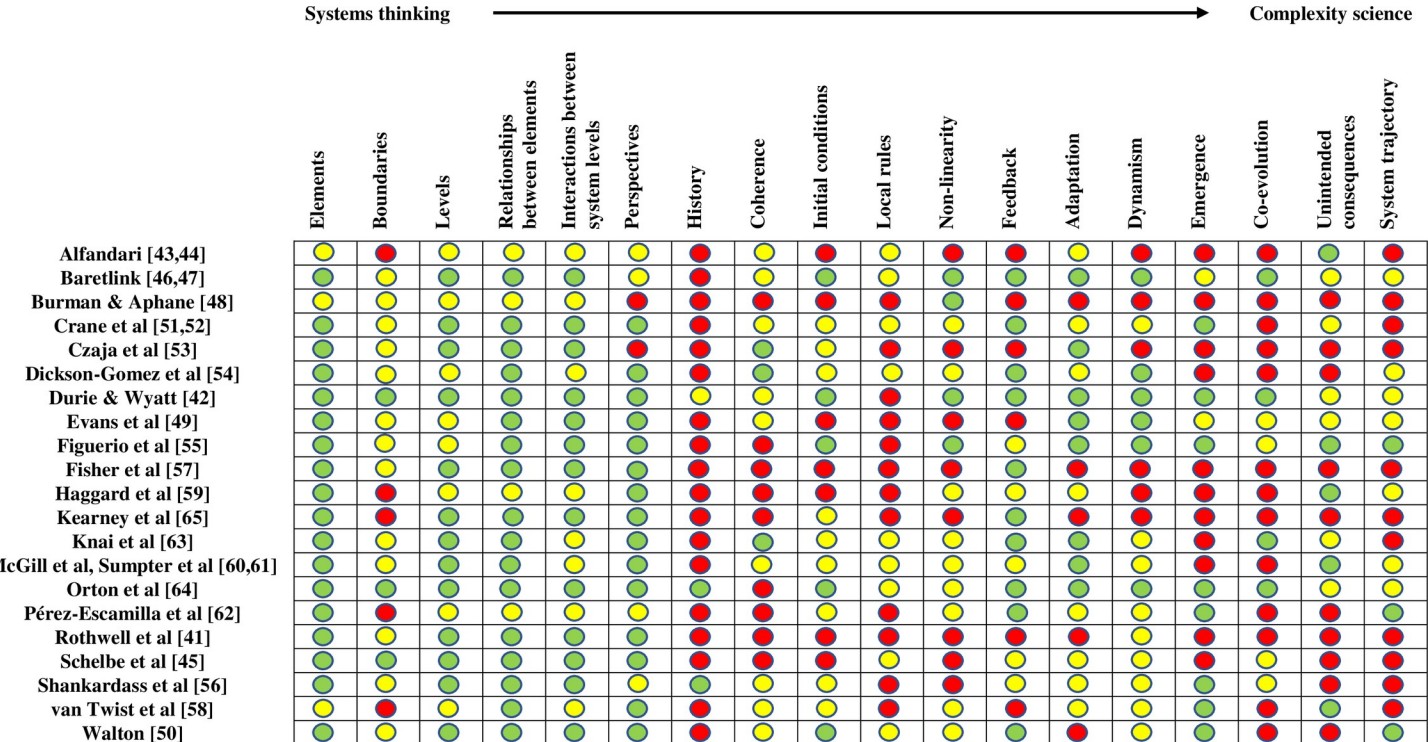

**Fig 2. Included studies and the degree to which they apply concepts from systems thinking and complexity science.** Each color-coded circle denotes the degree to which an evaluation applied the associated concept to any stage of the evaluation process. Green: study explicitly applied the concept; yellow: study attempted, or implicitly applied the concept; red: concept was not applied.

which, or indeed if, boundary judgments were made. By contrast, 11 studies produced some form of system diagram [41,46–48,51–57,60,63], implying that boundary judgments were likely at least implicitly considered by evaluators. The identified papers focused analytically on the relationships between systems elements. Such a focus is understandable and indeed, a prerequisite for being labeled as a system approach; without a focus on relationships and interactions—the key tenet of systems thinking—the approach fails to be systemic.

Somewhat surprisingly, only 4 fewer evaluations explicitly utilized a range of complexity concepts to assess changes within the system resulting from, or co-occurring with, intervention implementation over time [42,46,47,50,55]. By their nature, public health problems and the systems in which they are created and shaped are complex [40], and as a result, we might expect to see a more explicit attempt to use complexity concepts to generate evidence on public health interventions. Complexity science introduces a number of additional concepts that may be of value to researchers who seek to evaluate the mechanisms by which public health interventions have impacts in real-world environments. These concepts are used to describe, analyze, measure, and estimate attributes of change. The change first occurs within and across the system elements, and these collective changes result in emergent system change.

In the body of literature identified in this review, concepts from the complexity sciences, such as those that are used to understand change within systems, were utilized less frequently compared with concepts that could be used to describe static "snapshots" of systems. Although some papers were notable for applying a number of complexity concepts [42,46,47,50,55], the majority drew on only a few complexity-informed concepts in order to describe key mechanisms that might drive system change, such as a feedback loop. Researchers did not always

provide a rationale for how the concepts had been chosen or specifically considered within the context of data collection and analysis. An exception to this was one study that created an explicit analytic framework to identify and explain a range of by-effects (unintended consequences stemming from an intervention) [58]. The framework categorized policy achievements as foreseen or unforeseen and desired or undesired [58]. Within the evaluations identified, the complexity concepts that were most frequently used included nonlinearity, feedback, and adaptation.

## Discussion

We conducted a systematic search to identify examples of public health evaluations that apply a complex systems perspective to process evaluations involving qualitative methods. We then reviewed the systems and complexity concepts and methods currently used in this literature and found that evaluations of this nature draw on systems thinking to describe and analyze a system's structure at one point in time, whereas fewer draw on concepts from complexity science to assess change in a system over time.

We identified evaluations of a wide range of interventions affecting population health or their social determinants. These include interventions in school, workplace, and neighborhood settings in high- and middle-income countries, addressing behavior change, urban planning, community empowerment, health policy, and public–private partnerships. Public health process evaluations with a complex systems perspective have roots in a range of different disciplines and draw on a number of theories and frameworks to understand intervention implementation in real-world settings. The kinds of qualitative methods used in the included studies are in many ways similar to those founds in other (i.e., not focused on complex systems) forms of qualitative research: for example, in-depth and semi-structured interviews, focus groups, document review, and participatory methods. As such, the methods are not particularly novel, but rather, this body of literature is characterized by existing tools being paired with a complex systems perspective.

Half of the included studies produce some form of visual representation of the system they sought to describe. In most cases, these maps did not use formal system mapping techniques, and the diagrams varied greatly from study to study. Concepts associated with complex systems also seemed to be applied by many of the included studies in an ad hoc manner, rather than drawing from established theories and frameworks associated with the complex systems literature. Most studies claimed that their systems perspective was planned at the design stage of their evaluation, but few reported basing their approach around an established systems theory or framework [42,48,50,54]. Evaluators' attempts to utilize a complex systems perspective were most evident in the analysis stage of included studies, typically in the form of concepts from systems thinking and (less frequently) complexity science referred to in the analysis of qualitative data.

Included papers primarily utilized concepts from systems thinking to produce relatively static descriptions of systems and the interventions introduced within them. Although most evaluations concerned themselves to some degree with understanding mechanisms of, or barriers to, change, many did not make extensive use of the conceptual tools associated with complexity science that could help their attempts to better understand and unpack changes to the system of interest. In addition, although the evaluations identified in this body of literature drew on a range of qualitative methods, with many evaluators using a mix of qualitative methods within one evaluation design, it was often unclear why certain methods were chosen and the value added by each method.

From this summary of the review's main findings, we suggest that approaches to designing, conducting, and reporting qualitative process evaluations that have a complex systems

perspective are frequently underdeveloped and poorly specified. It is unclear to what extent systems thinking and complexity science influenced the key evaluation stages of study design, sampling, and data collection. The underlying theories informing evaluations are often unclear. The tendency to focus on systems concepts that describe a static system, rather than those best suited for assessing system change, seems counterintuitive, given that process evaluations are intended to assess mechanisms of change. We note that this rather critical assessment applies to many but not all of the studies we identified.

We would argue that all these studies are, in a sense, finding their way within an emerging field in which standards of best practice have yet to be established. We also believe that a contribution to the field would be a framework that seeks to address some of the problems identified in this review. Several authors have noted that although there are growing calls to utilize a complex systems approach, there have been fewer attempts to describe specific approaches or frameworks for doing so [35,71]. In particular, we advocate integrating a complex systems approach at the beginning of an evaluation design, to ensure that the perspective informs the evaluators' theoretical position, the evaluation focus, sampling strategy, data collection methods, analysis, and interpretation of findings.

In order to advance this area of public health evidence generation, we now consider some potential ways forward by proposing a framework for qualitative process evaluations from a complex systems perspective. Fig 3 shows our proposed evaluation framework, which involves 2 distinct phases. The first phase is intended to produce a static system description at an early time point. This is then followed by a second phase focused on analyzing how that system undergoes change. Specific steps in the evaluation are shown in the squares with directions and prompts to the evaluators at each step provided in italics. The figure underscores the ways in which the outputs of Phase 1 inform the direction and scope of inquiry during Phase 2. Table 2 also shows the role of qualitative methods in a process evaluation and how these map onto the application of concepts from systems thinking and complexity science.

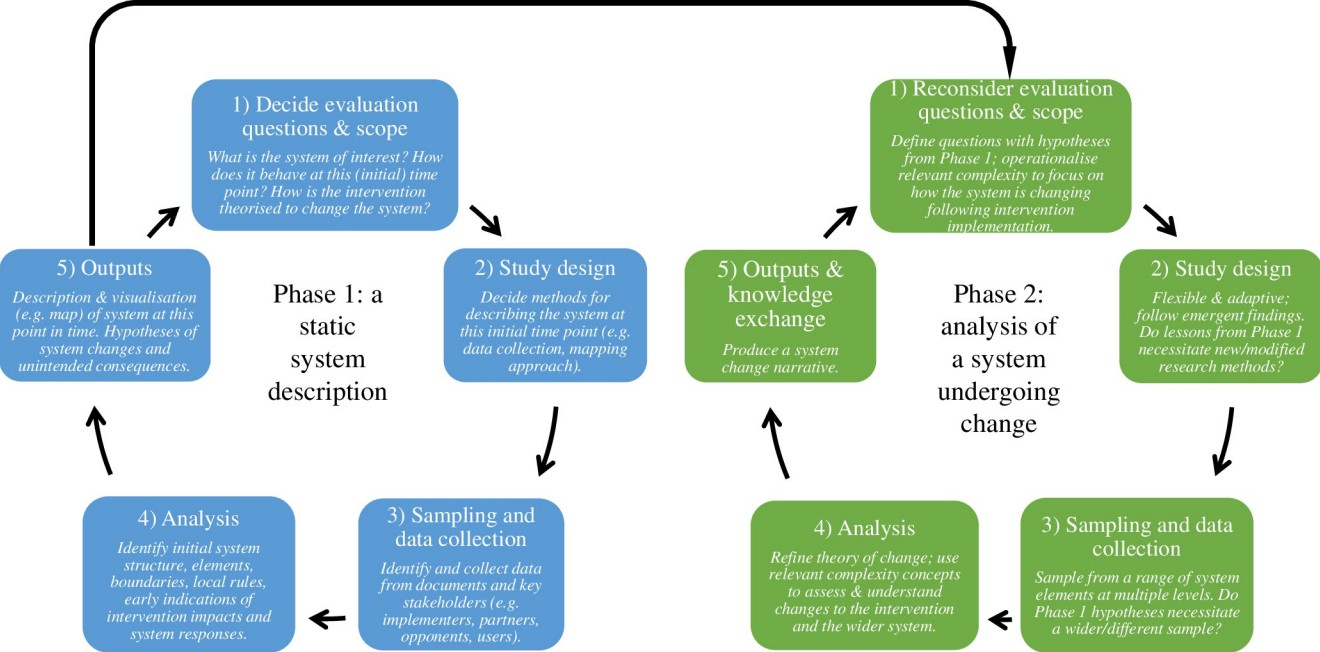

**Fig 3. Framework for a process evaluation from a complex systems perspective.** Evaluation stages are show in squares; the italicized font provides directions and prompts for evaluators at each stage.

**Table 2. Applying concepts from systems thinking and complexity science in a process evaluation.**

| | Concept | Definition | Process evaluation from a complex systems perspective | Methods |
|---|---|---|---|---|
| **Phase 1: A static system description (informed by systems thinking)** | Elements | Entities within a system, include, for example: people ("agents"), organizations, resources, etc. [12]. | Identify components of the system; begin a master list of system elements. | Concepts from systems thinking can be used to develop a **static system description**. A range of qualitative data generation methods are helpful to understand and produce a description of the system structure, including interviews, focus groups, workshops, (participant) observation and documentary analysis. For example, an evaluator could interview agents within the system to understand their views on the boundaries of the system, their role within the system and how their activities are influenced by other system elements, historical and contextual factors; observe a range of system activities to identify local rules and to assess coherence within the system; and conduct a documentary review (of intervention documents, relevant policies, reports, etc.) to understand the history of the system and to situate the system within its broader context. |
| | Boundaries | Decisions about what is included, and excluded in the system under observation; first-order judgments are boundary judgments made by actors within the system; second-order judgments are made by the evaluator [19]. | Assess first-order boundary judgments; combine primary data and evaluation considerations (e.g., scope of the evaluation, intended audience, pragmatic issues) to create "second-order" boundary judgment; create and revise system map as tool to guide boundary discussions, judgments and depiction. | |
| | Levels | A description of the structure of the system—may or may not be hierarchical [67]. | Describe the structure of a system. This can include identifying system levels (considering both vertical and horizontal dimensions) and exploring the ways in which system elements within and between levels relate and interact with one another. System structures and connections may be depicted in a (bounded) diagram. | |
| | Relationships | Connections or interactions between system elements [13]. | | |
| | Interactions | How system elements relate to each other and interact across system levels, or the broader context [14]. | | |
| | Perspectives | Different viewpoints of stakeholders within the system [18]. | Sample from a range of system elements; identify, assess, and report on a range of viewpoints. | |
| | History | The context before the initial conditions [68]. | Cast evaluative perspective beyond immediate system of inquiry and identify the broader context in which the system is located, as well as the context prior to intervention implementation. | |
| | Coherence | The extent to which elements' goals, activities and functions aligns with other another [69]. | Assess the degree to which system elements pursue the same goals and the ways in which their actions may promote or undermine each other's interests. | |
| | Initial conditions | How the system operates at "baseline"; these initial conditions set a system on a particular trajectory [24]. | Output of the initial stage of data collection and analysis; a relatively descriptive account that incorporates above concepts to depict the system of inquiry at a static point in time (often when an intervention is first implemented). | |
| | Local rules | The principles that guide interactions and behavior of system elements [14]. | Identify "if – then" statements or rules governing patterns of behavior in the system and of the system as a whole; use to understand and explain the ways in which interactions between system elements give rise to actions and behavior in the system. | |
| **Phase 2: Analysis of a system undergoing change (informed by complexity science)** | Nonlinearity | Inputs into the system do not necessarily result in correspondingly sized effects in the system; nonlinear relationships do not follow simple input-output line [40]. | Analyze interactions between systems elements to understand chains of cause and effect; define, draw and refine a theory of change which describe and depict the processes through which actions result in impacts, incorporating instances of feedback; evaluator may wish to draw causal-loop diagrams to visualize feedback loops. | Concepts from complexity science can be used to **analyze a system undergoing change**. Data collection will have a prospective element, with data generated longitudinally or at more than one time point in order to assess the ways in which the intervention and the system adapt and co-evolve with each other and the broader context. Qualitative data generation methods may include interviews, focus groups, workshops, (participant) observation and documentary analysis. These methods can be used to track changes over time and understand the processes by which change occurs. The data generated can be used to produce a narrative of the system undergoing change that underscores the factors that either amplify or dampen change; how the system and intervention adapt and evolve over time, any unintended consequences and how system elements' interactions generate emergent properties over time. Quantitative methods to measure impacts could include interrupted time series analyses, system dynamics modeling, agent-based modeling, network analysis. |
| | Feedback | Positive or negative response that may alter the intervention and its impacts. Positive feedback loops: change amplifies further change; negative feedback loops: change dampens down further change [6]. | | |
| | Adaptation | Adjustments in system behavior in response to internal and external change [6]. | Over a time period, both hone in on system elements and widen out evaluative gaze to system as a whole; ask "how do elements change their interactions with other system elements over time in response to the intervention?"; "how does the system change in response to the intervention?" "to what extent does the system absorb the intervention?" | |
| | Dynamism | Change in the state of the system that happens over time; time and evolution [7]. | Spend sufficient time in the field generating data to analyze system change over time; conceptualize both the system and evaluation as dynamic. | |
| | Emergent properties | Properties of a complex system that cannot be directly predicted from the elements within it and are more than just the sum of its parts; collective behaviors [70]. | Move evaluative focus from system elements to system as a whole and ask: "what types of system-level properties have emerged over time following the introduction of the intervention?"; explore system-level properties that cannot be attributed to individual elements. | |
| | Co-evolution | System change in response to its environment or another system; both systems change and evolve as a result [13]. | Look both vertically and horizontally; look at system elements and the system as a whole and ask: "in what ways does the system –and the environment it is in – change in response to the intervention?" | |
| | Unintended consequences | As a result of nonlinearity and feedback loops, complex systems are characterized by unanticipated processes and outcomes [22]. | Maintain an open stance and be open to unexpected impacts; follow-up on possible impacts that may not feature in the original theory of change. | |
| | System trajectories | Includes path dependency [68], attractor state [12], phase space [68], phase transition [24] and bifurcation/tipping points [14]. Qualitatively, the path a system follows through time, moving through different states, including periods of stability and instability [7]. | Narrative of a system undergoing change; output of data analysis is a "system story" that incorporates concepts from systems thinking and complexity science. | |

## Phase 1: A static system description

In the first part of this 2-phase framework, we propose that evaluators conduct a period of research in order to gain an initial understanding of the system, including the system structure, the boundaries, the constituent elements, and the relationships between these [6,14] at a given time point [24]. This description represents a snapshot of the system at one point in time. For many evaluators, it may make sense to capture the "initial conditions" or "initial state" of the system at the time the intervention is first implemented. In these cases, the evaluation would involve a period of familiarization and the first part of data collection as the intervention is being implemented or shortly thereafter. In this stage, evaluators would also begin to hypothesize some of the ways that the intervention may lead to change within the system (which may be informed by the intervention's theory of change, if one is articulated). If the intervention designers have not described a theory of change, evaluators at this stage should articulate one by mapping out the initial hypotheses of system change.

In Phase 1, evaluators would begin to make sense of and document the "local rules" that govern both the intervention and the system, including the rules that govern how different system elements interact and relate to each other and how the intervention operates and relates to different parts of the system. In undertaking Phase 1, evaluators would draw on concepts that are most closely aligned with systems thinking (the left-hand side of Fig 2 and first half of Table 2) and use these to structure the initial data collection and analysis. Following the identification of the system structure, elements, boundaries, and relationships, evaluators should begin to consider some of the ways in which the intervention may lead to changes within the system. Evaluators could ask how the system elements respond to the intervention, comparing different stakeholder perspectives. Evaluators could also begin to assess system coherence by analyzing the degree to which the intervention is aligned with the interests of those in the system or the instances in which the intervention may "swim against the tide" [72,73].

In Phase 1, data should be collected from a range of different actors within the system. Evaluators may find a number of different data collection methods useful, including, but not limited to, an initial documentary review, interviews, and workshops. The boundary decision and the identification of system elements will inform from whom data are collected and through which methods [14].

As part of this process and as a way of analyzing the data collected in Phase 1, it may be helpful to create a map of the system. The type of map created will depend on the role it is to play in the evaluation. For example, if a map is made to visually represent the system structure and boundaries to help depict and understand the system structure and relationships between the system elements [57], it may be created through a semi-structured brainstorming session or interviews and the analysis of the data collected in Phase 1. Alternatively, evaluators may choose to create more structured system maps, drawing on established mapping methods, such as concept mapping or group model building, in order to map out causal linkages between system variables [74]. In these instances, Phase 1 represents an opportunity for initial preparatory work for the map creation process.

The output of Phase 1 would be relatively descriptive and static: a qualitative description of the system structure, elements, boundaries, and relationships which may well be depicted on a map, as well as some hypotheses about how the intervention may lead to system change, including the ways in which the elements and the system as a whole adapt and co-evolve in response. The hypotheses of system change may be depicted as a theory of change, which maps out how the intervention could lead to impacts, with particular consideration given to the pathways and mechanisms by which that change is brought about [6]. The initial system description and possible pathways for system change would then inform Phase 2.

## Phase 2: A system undergoing change

The second phase of evaluation would examine emergent properties of the system and explore system change stemming from the intervention, drawing on a complexity perspective. In Phase 2, evaluators should be prepared to follow the pathway of emergent findings. In this sense, the evaluation needs to be adaptable, flexible, agile, incorporate multiple perspectives, and deal with uncertainty to support real-time decision-making. Evaluators would use the data collected in Phase 1 (particularly the emerging hypotheses about system change) to develop specific research questions about the intervention and the system. In defining the research questions, there is an opportunity to explicitly apply some of the complexity concepts—for example, by asking questions about the adaptive responses within different elements of the system, unintended consequences of the intervention for different population groups, or emergent system outcomes as the system co-evolves with its broader environment. It is not our suggestion that evaluators attempt to apply all complexity concepts to any one evaluation but rather focus on those that can generate useful evidence for decision-making [71]. Although the timing of Phase 2 may be determined by the theory of change, it may also be influenced by the timing of other types of data collection. For example, the process evaluation may accompany an impact evaluation that prespecifies time points for data collection [16,17].

At this stage, a more formal period of sampling and data collection would begin, to complement data collected in Phase 1 and to focus the sampling and data collection strategies to better answer the research questions. The specific sampling strategy and data collection methods will vary from evaluation to evaluation, but any process evaluation applying a complex systems perspective would sample multiple types of participants (e.g., different system elements) and use multiple methods [6,66]. As the papers in this review underscore, the careful use and reporting of different qualitative methods underpinned by complex systems theoretical principles can help an evaluator assess different perspectives across and within system levels, as well as different types of information [27]. Analyzing data generated through different qualitative methods can be used to bring a dynamic component to the evaluative research; for example, documents can be used to understand previous decisions and interviews or observations could then be used to understand the trajectory of those decisions and their impact across the system on different population groups [27]. Evaluators should consider the timing and ordering of mixed methods; a document review might, for example, provide important context in order to inform interview schedules [27]. Complexity concepts have traditionally been used within the context of quantitative and modeling methods. However, we argue that there is no reason that these concepts should not be of interest within a process evaluation using qualitative methods, particularly as many deal specifically with system changes upon which qualitative research could shed light [41,48].

During the analysis stage, the evaluators would begin to make sense of the emerging findings through the application of relevant complexity concepts. For example, an evaluation concerned with understanding the ways in which the intervention may lead to the amplification or dampening down of certain kinds of systemic change would have an explicit focus on identifying feedback loops within the system [75], or it might make sense (based on hypotheses generated in Phase 1) to focus the analysis on understanding how the system's history influences its trajectory and adaption in response to the introduction of an intervention [76]. As the analysis is undertaken, there is likely a need to collect more data, in a kind of evaluative feedback loop. Such a process will be familiar to those who apply iterative research designs [17,77]. Throughout the analysis, evaluators would revisit, revise, and refine the theory of change and system map in light of the new data.

Generating outputs can be a challenge for public health evaluators applying a systems perspective. It is difficult to convey complex findings in a manner that is useful and timely for

decision makers and does not result in an overly reductionist account or a confusingly "complex" set of findings. This is particularly a concern for qualitative research in which large volumes of data are collected. We suggest that one way to present the findings from a complex systems process evaluation is to create a "system story," wherein the evaluator describes and analyses how the intervention embeds and co-evolves with the system and its elements overtime [3].

A more traditional approach to process evaluation is often rooted in the intervention itself, rather than the system in which that intervention is implemented. As a result of this orientation, such an evaluation generally considers the intervention and its immediate implementation processes and mechanisms, although there may be some consideration of more distal mechanisms and impacts [17]. In addition, more traditional process evaluations tend to adhere to research protocols that may themselves be relatively inflexible. A process evaluation from a complex systems perspective takes the system as the initial starting point of the analysis and considers the ways in which the intervention may lead to immediate, as well as more distal impacts, and the ways in which that intervention may change how the system elements—and the system as a whole—behave. Doing so will inherently require a flexible, adaptive, and iterative design. The framework presented here suggests at least 2 phases of data collection, with the understanding that the second phase will likely include an iterative process of defining research questions and collecting and analyzing data. Utilizing a longitudinal design with data collected over a relatively lengthy period of time or at more than one time point in order to capture a dynamic system undergoing change [24,67,71] may be a challenge to public health evaluators because it implies longer timescales [78], a move away from more standard evaluative approaches and a degree of risk with which some funders and decision makers may be uncomfortable. In addition, it may challenge traditional public health evaluation methods that strictly follow protocols in an attempt to control for internal validity [16]. In contrast, a complex systems approach to evaluation must inherently plan to adapt and change in response to early evaluative findings, as well as in response to the changing intervention and broader system. As a result of an adaptive evaluation design, the distinction between different types of evaluation (such as formative, process, outcome, and impact) may be less clearly defined. As evaluators follow the pathways of emergent hypotheses and findings, it may well make sense to, for example, measure or predict impacts alongside process mechanisms. Finally, further work remains on the ways in which realist and mixed methods approaches can more explicitly contribute to a process evaluation from a complex systems perspective, but it is beyond the scope of this current review.

## Limitations

The nature of the review topic area required the research team to make a number of judgments throughout the review process. First, judgments were made regarding which studies to include or exclude on the basis of their public health relevance and the degree to which they featured a complex systems perspective. Although the majority of decisions were clear cut, the reviewers, in discussion with one another, had to make judgments in cases that were less obvious, and there is the possibility that other review teams would have made different decisions. In addition, there was a subjective element in deciding which concepts from systems thinking and complexity science to highlight; we sought to capture the key principles associated with each of the traditions with the goal of this list being used by those wishing to draw on systems thinking and complexity science within the context of public health evaluation. We recognize that other reviewers might have chosen to highlight other concepts. Finally, the critical appraisal of the studies again required judgments. In order to increase validity, 2 reviewers completed the process independently and reconciled their decisions, but the decisions were not always clear cut.

Another limitation of this review is the focus on studies which self-identify as taking a systems and/or complexity-informed approach. This focus has 2 possible limitations: First, it excludes studies that may be compatible with systems thinking but do not cite systems literature or draw explicitly on systems concepts, and second, it may include studies that utilize the terminology of complex systems, because it has become somewhat fashionable in the last few years, but fail to apply the concepts in such a manner that investigates complex uncertainties to generate better evidence for decision-making [71]. Taking the first concern, many rigorous qualitative studies foreground context in their research focus and analyses, considering the broader economic, social, political, cultural, environmental, and historical factors that impact interventions' trajectories and influence diverse population groups [79]. As we have contended, "system" and "context" are broadly synonymous, in that all of a system can arguably be considered "contextual." Therefore, qualitative research that actively engages with the broader context may apply a perspective that is compatible with systems thinking, without using the accompanying systems terminology. Indeed, the MRC Guidance on "Process Evaluation of Complex Interventions," had limited reference to complex systems theory and terminology but nevertheless advocated a systems-compatible approach to process evaluation, namely, an approach that explores the "dynamic relationships between implementation, mechanisms and context, the importance of understanding the temporally situated nature of process data in understanding the evolution of an intervention within its system" [17,71]. With regards to the second concern, complex systems thinking is currently in vogue in public health, which can be seen in the growth of calls for the application of a complex systems perspective to public health practice and research [1,35,80,81]. Although many researchers are grappling with how to harness insights from the systems thinking and complexity science traditions to improve public health research, there is some concern that complex systems literature and concepts have been used without researchers truly engaging with the underlying theory [71]. These limitations suggest a number of opportunities for further research in this field. In particular, future research could fruitfully explore the degree to which public health literature—on intervention development and evaluation—is compatible with a complex systems perspective, even when not explicitly described as such. Other research might identify process evaluations that do not explicitly adopt a complex systems approach and analyze the added value of an explicit engagement with the systems and complexity literature.

Finally, we limited our search to English-language publications and relied on 2 previous reviews and an expert consultation to identify qualitative process evaluations from a complex systems perspective that were published prior to 2014, which is a limitation of our search's sensitivity. The studies identified through these means may have been influenced by other researchers' interpretations and possible biases. Any papers not identified from our search may have potentially added further to our methodological synthesis and the recommendations we put forward in the Discussion.

## Conclusions

We have conducted a systematic review to identify qualitative process evaluations of public health interventions that consider themselves to be informed by systems thinking and/or complexity science, and we have analyzed the extent to which they feature key concepts from these fields. We found that this area of public health evidence generation is still in early stages of development and there is little consensus on a general approach. Informed by our evidence synthesis, we have therefore developed a framework for process evaluations that assesses change within the context of a wider complex adaptive system. We suggest that to do this, evaluations themselves need to be designed with a complex systems perspective, which requires

being agile and adaptable in order to capture the system change they seek to assess. We are currently testing out this approach in an evaluation of how a system and its elements adapt and co-evolve in response to a local alcohol intervention that raises additional revenue to police and manage the night-time economy. We intend that this 2-phase framework can be of use, and be further refined, by public health practitioners and researchers who seek to produce evidence to improve health in complex social settings.

## Supporting information

**S1 PRISMA Checklist. PRISMA, Preferred reporting items for systematic reviews and meta-analyses.**
(DOC)

**S1 Text. Example search strategy.**
(DOCX)

**S2 Text. Case study examples from the systems thinking and complexity science traditions.**
(DOCX)

## Acknowledgments

We thank the wider research team who have worked on the National Institute for Health Research, School for Public Health Research (NIHR SPHR) project "Developing a systems perspective for the evaluation of local public health interventions: theory, methods and practice." Rachel Anderson de Cuevas (University of Liverpool), Steven Cummins (London School of Hygiene & Tropical Medicine), Frank de Vocht (University of Bristol), Karen Lock (London School of Hygiene & Tropical Medicine), Petra Meier (University of Sheffield), Lois Orton (University of Liverpool), Jennie Popay (Lancaster University), Harry Rutter (University of Bath), Natalie Savona (London School of Hygiene & Tropical Medicine), Richard Smith (University of Exeter), Margaret Whitehead (University of Liverpool), and Martin White (University of Cambridge) commented on the search terms and/or helped identify potentially relevant studies. In addition, we thank those academics who responded to our expert consultation; these include Zaid Chalabi (London School of Hygiene & Tropical Medicine), Peter Craig (University of Glasgow), Seanna Davidson (The Australian Prevention Partnership Centre), Ana Diez Roux (Drexel University), Anna Dowrick (Queen Mary University of London), Diane Finegood (Simon Fraser University), Penny Hawe (The University of Sydney), Vittal Katikireddi (University of Glasgow), Laurence Moore (University of Glasgow), David Peters (Johns Hopkins University), Mat Walton (Massey University), and Katrina Wyatt (University of Exeter).

## Author Contributions

**Conceptualization:** Elizabeth McGill.

**Data curation:** Elizabeth McGill, Vanessa Er, Tarra Penney.

**Formal analysis:** Elizabeth McGill, Dalya Marks, Vanessa Er, Tarra Penney, Matt Egan.

**Funding acquisition:** Mark Petticrew, Matt Egan.

**Investigation:** Elizabeth McGill.

**Methodology:** Elizabeth McGill, Dalya Marks, Mark Petticrew, Matt Egan.

**Project administration:** Elizabeth McGill.

**Resources:** Elizabeth McGill.

**Software:** Elizabeth McGill.

**Supervision:** Dalya Marks, Mark Petticrew, Matt Egan.

**Validation:** Elizabeth McGill, Dalya Marks, Matt Egan.

**Visualization:** Elizabeth McGill.

**Writing – original draft:** Elizabeth McGill.

**Writing – review & editing:** Elizabeth McGill, Dalya Marks, Vanessa Er, Tarra Penney, Mark Petticrew, Matt Egan.

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
