## [Editor Report · Decision Letter 0]

15 Jan 2020

Dear Dr McGill, 

Thank you for submitting your manuscript entitled "Process evaluation with a complex system lens: a systematic review and framework for public health evaluators" for consideration by PLOS Medicine.

Your manuscript has now been evaluated by the PLOS Medicine editorial staff and I am writing to let you know that we would like to send your submission out for external peer review.

**Please be aware that, due to the voluntary nature of our reviewers and academic editors, manuscript assessment may be subject to delays during the holiday season. Thank you for your patience.**

Kind regards,

Helen Howard, for Clare Stone PhD

Acting Editor-in-Chief

PLOS Medicine

plosmedicine.org

---

## [Decision Letter · Decision Letter 1]

11 Feb 2020

Dear Dr. McGill,

Thank you very much for submitting your manuscript "Process evaluation with a complex system lens: a systematic review and framework for public health evaluators" (PMEDICINE-D-20-00099R1) for consideration at PLOS Medicine. 

[LINK]

In light of these reviews, I am afraid that we will not be able to accept the manuscript for publication in the journal in its current form, but we would like to consider a revised version that addresses the reviewers' and editors' comments. Obviously we cannot make any decision about publication until we have seen the revised manuscript and your response, and we plan to seek re-review by one or more of the reviewers. 

We expect to receive your revised manuscript by Mar 03 2020 11:59PM. Please email us (plosmedicine@plos.org) if you have any questions or concerns.

We look forward to receiving your revised manuscript. 

Sincerely,

Caitlin Moyer, Ph.D.

Associate Editor 

PLOS Medicine

plosmedicine.org

Abstract – please include a sentence on the limitations of the study as the final sentence of the ‘Methods and Findings’ section.

Please avoid jargon (e.g., "with a complex system lens"; "relatively static descriptions of the system under inquiry") and explain more clearly what you found and what it might mean.

Are you able to include the names of the "32 academics"?

Please use sections and paragraphs instead of page numbers in the checklist – as these can change during revisions / formatting etc. 

You say in the PRISMA that the protocol is not in the public domain yes in the submission form in relation to data you say ‘This is a review of published studies; all studies are in the public domain. I would have thought if all studies are in the public domain the protocol would be too. Please explain. 

Comments from the reviewers:

Reviewer #1: I was asked to provide a statistical review of this paper. However, no statistics were used, so I have no comments on them

Peter Flom

Reviewer #2: 

This is a well-written, clear and very useful paper. It provides a nice conceptualization of complex systems thinking, and of static versus dynamic perspectives. The glossary and list of examples will be invaluable to persons working in this field. 

My only substantive comment relates to the presentation of the 22 processes evaluations from the systematic review. These are well presented in the table and at a high-level in the main text and in Figure 2. They form a very useful repository of examples. However, short of digging out some of the references, I didn't come away with a clear picture of any exemplar of an evaluation that used systems thinking or complexity science. Perhaps due to word count it is not possible to present 1-2 examples in more detail: I leave that with the authors and editor to consider.

A few very minor comments.

Page 19, glossary, "coherence": "… functions aligns with other another". Please correct.

Page 18, "analytical framework" I believe should read "analytic framework".

Page 23, should read "data are collected…".

Page 26, Conclusions: Might the first sentence of this section better read "… informed by systems thinking and/OR complexity science.."? 

Page 27, could you replace "evaluate" with another verb (eg assess) "we have therefor developed a framework for process evaluations that evaluate…"?

Page 27, might delete "ourselves", to read "we intend to test out this approach in the near future.".

Reviewer #3: Thank you very much for providing me with the opportunity to review this interesting article on process evaluations with a complex system lens. It is the intention of the authors to develop a framework for researchers (?) who are planning the evaluation of a complex public health intervention. 

While the scope of the manuscript is really interesting, I think the manuscript would benefit from some revisions. 

Title: 

- Maybe you should integrate the focus on qualitative methods into the title, depending on how much you want to emphasize this component

Abstract

- It does not become quite clear why complex interventions call for the application of qualitative methods; please formulate more precisely

- In the result section, you talk about analytical approach; not sure whether I would rather talk about conceptual approach or even underlying theory which is weaved into the entire process; is it systems concepts or system concepts? The last sentence of the result section is really hard to understand; can you be more precise?

- Conclusion: 

o is it that methodological principles are underdeveloped? Or is it rather that methods (or even methodologies) do not justice to existing theories or conceptualizations of complexity? I think it is a great disconnect in public health literature with regards to the conceptualization of complexity and the methods (or methodologies) which are chosen to address those 

o Concepts that should be operationalized: again, I think you need to be more specific

o I am not sure whether the focus should be on qualitative methods only; maybe it is more about integrating qualitative methods into the assessment of complex interventions alongside other methods. Maybe use the term "Integrate"

Background

- I don't think you need the subheadings; I suggest rather to focus on connecting the concepts also in the background section

- Complexity: Maybe you can work on connecting the complex system and the complex intervention paragraph more closely; I would include the conceptualization of complex interventions as being considered events in complex systems, stress the interplay between the system and the intervention and that one can hardly be considered without the other.

- Evaluation: I think you need to specify other types of evaluations, such as outcome/impact evaluations; process evaluation is usually conducted alongside these evaluations; please provide rationales for each type of evaluation; 

- Process evaluation definition: could you provide a definition of process evaluation? (e.g. Moore et al. 2015)

Objectives:

- Please be specific (also in the abstract) about what process evaluation approaches you are looking at: purely qualitative ones or the ones integrating qualitative approaches

- The three goals are a bit misleading considering your objective; you were not looking for "types of public health interventions that have been subjected to process evaluations using a complex system lens" but the only the ones that have been assessed with (among others?) qualitative methods; be more precise about your objectives; also, the conceptualization or theory of complexity underlying the respective process evaluation approach was probably of interest to you.

- Be also more specific about the framework you aim to develop; you can refer to Nilsen et al (2015) for a terminology of framework in Implementation Science. Also add the intended audience to the objectives.

Methods

- Why did you choose Jan 2014 as starting point? Please provide rationale

- Add language limitation to the text

- I am a bit surprised that your search strategy does not contain any search terms for qualitative methods; after having started reading the article in the abstract and the objectives, one would assume that this was a primary goal of the inquiry

- You should provide more details on your inclusion criteria; this is a conceptual review which means that including and excluding articles can become quite a challenge; it is therefore even more important that you provide us with your exact definition of each inclusion criteria, namely

o Self-identified system approach: where would that happen? In the background, method or discussion section? If you only use it to embed your findings, I don't assume this would be sufficient

o Relevant to public health: I am not sure how to interpret that; was that a subjective judgement? How did you define relevance? How did you define public health? I could imagine quite some grey areas around this

o Process evaluations: did they have to be process evaluations only? What if it was an outcome evaluation alongside a process evaluation? Would that also be included? Again, also provide your definition and potential criteria you used

o Qualitative methods: which definition did you use? I am wondering if methods such as document analysis etc. would be covered by your understanding of qualitative methods; also, would the process evaluation use ONLY qualitative methods, or would studies using mixed methods also be comprised? If so, please also provide your understanding of that (e.g. weighing of methods, sequence of methods etc. cp. Creswell et al.)

- List of concepts of complexity science and system thinking: 

o it seems like this is where you lay out your understanding of complexity science and system thinking; you are considering them as two sides of one spectrum; I am wondering what spectrum this is and how "simplicity" fits in? 

o How was this list developed? Why were certain aspects chosen and others not?

o What is the difference between context, history and initial conditions? aren't local rules also context? 

o What is elements? How does it relate to context and levels?

o Dynamics and interactions also seem to have some conceptual overlaps; 

o Summary: Did you take a look at other conceptualizations of complexity, such as https://www.ncbi.nlm.nih.gov/pubmed/28446138 ? It seems that there is quite an overlap between some of these concepts; it is hard to understand the rationale for some of these concepts and their implication for evaluation

- I would call the section on quality appraisal something like critical appraisal. Could you be more specific about the ratings? At what stage would any of these concepts be applied? Would it be important to know whether a "complexity lens" was applied at the conceptualization or design stage? Would this make a difference compared to applying it at the analysis/interpretation stage when all data has already been collected?

- The understanding of "application" of a "complexity lens" seems quite fuzzy to me; Can you be more specific about both terms?

- Throughout the analysis/synthesis section, I am not quite sure how the qualitative methods aspects comes in; how were the two concepts - complexity and QRM - connected? 

- What I think is missing is the actual method extraction: what was the mode of evaluation, which time points, which methods at which time point, at what stage complexity lenses have been applied and so on; this could also be integrated into the result section; just think about your audience: what would a person who wants to evaluate complex interventions want to know? 

Results

- What is public health strategy? It does seem to fall out of line with the other fields of application

- Disciplinary background? I assumed the disciplinary background was public health? Or are you referring to the primary authors?

- Studies drew from… - with regards to what? 

- Can you provide an overview of all complexity theories applied?

- Can you provide an overview over the stages at which the frameworks or theories have been applied? 

- Were the frameworks you report only used for structuring the analysis or already at the planning stage?

- From a researcher' point of view, I think it would be interesting for what purpose one or more of the qualitative methods have been employed; what was their sequence? How did they inform other components of the evaluation? E.g. document analysis becomes more and more comment, but is used for different purposes; at what stages were they used (baseline, interim, final)? Did the purpose differ?

Discussion

- Moving from a review to guidance is quite a normative process; I am not entirely sure whether I would put this into the discussion or the result section; if put into the result section, it should be made clear how this framework came about; as an alternative, the framework could represent a summary of current practice, with each of the steps being informed by included studies; you could use this framework as a starting point and check which of the included studies followed this process. Maybe there is a step that you have missed? You could then discuss potential missing steps/aspects in the discussion

- Use of theory of change or logic model should be strongly encouraged

- Qualitative research usually has a strong theoretical underpinning; how does complexity theory align with other theoretical perspectives (e.g. poststructuralism)? How does complexity theory augment qualitative research methods?

- I wonder how the respective "complexity lens" and the chosen qualitative research methods relate to each other? Were they chosen based on the lens or was it rather a purposive choice?

- In order to claim a complexity perspective, I think all of the evaluations need to have a complexity perspective from the very beginning; I didn't go through the primary studies, but maybe this is something worth taking a second look at

- What you should also take another look at is the overall development, implementation and evaluation of a complex PH intervention process; do qualitative methods only come in at the evaluation stage? When does evaluation start? 

Table 1: 

- Please specify roots; and also be consistent with the text

- Application of complex systems: What does that mean? Please be more specific

- System map: what is this? Please also put into text!

- Could you describe the types of evaluative approaches that have been undertaken unless all of them are process evaluations only? Were they all continuous? How many points of measurements did they have? And did each of the measurement points comprise the same methods? 

- I would also put in the stage at which a complexity lens was used

- Maybe also add whether or not a logic model was used (quite standard in complex interventions)

Flow Chart: 

- you excluded studies that were not primarily qualitative, but then the result sections reports mixed method studies? How does that fit together?

- Not public health? How does that fit with the disciplinary background?

- What does "not empirical finding" mean? Why has that not been included in the inclusion criteria? 

Figure 2: you could consider using the Cochrane RoB display as alternative to this graphic display; it is a bit easier to understand

Reviewer #4: This is a very interesting and well written article, which I think makes an important contribution to the methodological literature around process evaluation and systems thinking.

I have a two quite minor and general comments regarding some of it's framing, limitations and contribution to the literature which the authors could consider.

First, it would be good to see some more extended reflection in the discussion about the limitations of focusing on studies which self-identify as using systems perspectives. Because the focus is limited to qualitative studies, systems thinking is likely to be more prevalent than complexity science. However, as the authors set out, this is a way of thinking and seeing the world more than a discipline or method. Hence, it is perfectly possible for authors in deciding their research questions and planning their studies to be guided by ways of thinking which are compatible with systems thinking, without ever using that term and citing the systems literature. In fact, many elements of systems thinking can probably be retrospectively identified within most good quality qualitative research. My sense is that what the field is trying to do at the moment, which in itself is a major attempt to disrupt the evidence production system, is to normalise systems thinking in evaluative practice such that it no longer becomes a separate bounded approach to doing evaluation and just becomes what people do. For many in this field, that is perhaps already the case. On the other hand, during this period of system disruption, lots of people began to use the term systems thinking in the past 5-10 years without really engaging with its core concepts, just because it has become something of a fashionable buzz-phrase and they thought they needed to say they were doing it to get funded and published. So I think a useful next phase to take this work further would be to identify and compare qualitative or mixed method process evaluations which do or do not explicitly cite adopting a systems framework, and delineate what value if any is added by that more explicit engagement with the systems literature, and how much systems thinking compatibility is emerging in studies which don't assign that label to themselves explicitly. 

I think there could be a little more recognition that many of the issues being discussed here were present within 2014 MRC guidance for process evaluation, which did pay attention to the need for flexibility to respond to emerging issues, and explicitly put context front and centre as the largest box within the diagram, with pre-existing contextual/system conditions framing everything that followed in terms of what kinds of intervention were selected, what mechanisms they activated and what outcomes occurred as a consequence. It maybe shied away from bringing a complex systems perspective too much to the forefront due to a desire to take people along with it, rather than leaving people behind by being too radical. But I understand the forthcoming MRC guidance which we should expect to see sometime in 2020 will be more bold in that regard, reflecting how the evidence production system has shifted over time. More recently, this article has revisited some of the systems-consistent elements of this guidance (Moore, Graham, Evans, Rhiannon, Hawkins, Jemma, Littlecott, Hannah, Melendez-Torres, Gerardo, Bonell, Chris and Murphy, Simon 2019. From complex social interventions to interventions in complex social systems: future directions and unresolved questions for intervention development and evaluation. Evaluation 25 (1) , pp. 23-45. https://journals.sagepub.com/doi/10.1177/1356389018803219) and actually I think there is a lot of agreement between the recommendations of your review and new framework, and some of the statements within that article around the need to understand interventions explicitly as attempts to change how systems function, starting by understanding the system before focusing on how a new way of working alters it and what consequences that produces etc...

Graham Moore

[LINK]

---

## [Decision Letter · Decision Letter 2]

25 May 2020

Dear Dr. McGill,

Thank you very much for submitting your revised manuscript "Qualitative process evaluation with a complex systems perspective: a systematic review and framework for public health evaluators" (PMEDICINE-D-20-00099R2) for consideration at PLOS Medicine. 

Your paper was evaluated by a senior editor and discussed among all the editors here. It was also sent to one of the original reviewers, as well as an additional methodological reviewer. The reviews are appended at the bottom of this email and any accompanying reviewer attachments can be seen via the link below:

[LINK]

In light of these reviews, I am afraid that we still will not be able to accept the manuscript for publication in the journal in its current form, but we would like to consider a revised version that addresses the reviewers' and editors' comments. Obviously we cannot make any decision about publication until we have seen the revised manuscript and your response, and we plan to seek re-review by one or more of the reviewers. 

We expect to receive your revised manuscript by Jun 15 2020 11:59PM. Please email us (plosmedicine@plos.org) if you have any questions or concerns.

We look forward to receiving your revised manuscript. 

Sincerely,

Thomas McBride, PhD

Senior Editor 

PLOS Medicine

plosmedicine.org

1- Abstract Background: perhaps there is an alternative to using “review” twice in the same sentence.

2- Please ensure that all references use the "Vancouver" style for formatting, and see our website for other reference guidelines https://journals.plos.org/plosmedicine/s/submission-guidelines#loc-references

3- Please remove the contraction from the 1st point of the Author Summary.

4- Please add “to our knowledge” or similar to the 4th point of the Author Summary

5- Author Summary, 7th point: please be more specific than “a lot”.

6- Please rename the “Background” section “Introduction”.

7- Please describe how the 32 academics were identified.

Comments from the reviewers:

Reviewer #3: Thank you for providing me with the opportunity to review this manuscript once more - after having been particularly picky around the methods. My apologies, this only happens when I find things particularly interesting and really think about what implications this has on our research practice. 

I went through the comments provided by other reviewers and re-read the manuscript. I think the manuscript improved very much and I would like to acknowledge the efforts that went into the revision of the manuscript. In particular I enjoyed reading the authors reflections surrounding the role of complexity theory in paradigm discussions in QRM. 

I have three minor comments:

- I appreciate the efforts that went into revising the methods section; by specifying your inclusion criteria and acknowledging the many grey areas around the concepts which determined inclusion or exclusion, I think you already contribute to moving the discussion forward; I would still encourage the authors to link their conceptual work to existing conceptual work around public health, qualitative research methods and mixed methods (e.g. substantial component is quite subjective; there is however literature which describes the role of qualitative research in mixed method which you could use as a basis to further specify your inclusion criteria. If you feel that this is too much text, you could also do this in tabular form.

- Page 13: maybe use "yellow" rather than "amber", cause it might be easier to understand.

- "The aim of this analysis was not to be overly critical about individual studies, but rather to understand the ways in which concepts from systems thinking and complexity science are applied in this body of literature"  maybe substitute analysis by appraisal

I am looking forward to seeing this published and taken up by the research community.

Reviewer #4: I very much enjoyed reading this paper, and I think it represents a very important and timely methodological contribution

Reviewer #5: See attachment

Michael Dewey

[LINK]

---

## [Decision Letter · Decision Letter 3]

12 Aug 2020

Dear Dr. McGill,

Thank you very much for re-submitting your manuscript "Qualitative process evaluation from a complex systems perspective: a systematic review and framework for public health evaluators" (PMEDICINE-D-20-00099R3) for review by PLOS Medicine.

I have discussed the paper with my colleagues and the academic editor and it was also seen again by the statistical reviewer. I am pleased to say that provided the remaining editorial and production issues are dealt with we are planning to accept the paper for publication in the journal.

[LINK]

We look forward to receiving the revised manuscript by Aug 19 2020 11:59PM. 

Sincerely,

Thomas McBride, PhD

Senior Editor 

PLOS Medicine

plosmedicine.org

Requests from Editors:

1- In the Abstract Methods and Findings, instead of "We identified examples ..." please note that you did a systematic search and include the search dates.

2- Though it may not be possible to list all of the interventions, perhaps the Abstract could provide examples to demonstrate the “ wide range of public health interventions”. Also useful would be a description of the range of settings, number of different countries, and the breakdown of high/middle/low-income settings.

3- Again in the Abstract, rather than "Qualitative researchers lack a common understanding ..." , we suggest you say what you found instead

4- In the Abstract Methods and Findings and the Results, please quantify the main outcomes, rather than “Fewer” or “Most”.

5- Please remove the subheading “This review” from the Introduction.

6- Is it possible to list the 32 academics contacted?

7- Thank you for including your PRISMA checklist. Please add the following statement, or similar, early in the Methods: "This study is reported as per the Preferred Reporting Items for Systematic Reviews and Meta-Analyses (PRISMA) guideline (S1_PRISMA_Checklist)." Please also replace the page numbers for the first two items and refer to the sections (Title and Abstract).

8- In the main text, please include a legend for Figure 2 below the figure title. Similarly for Figure 3.

9- The last paragraph of the Results section “In summary…” seems more suited to the Discussion section.

10- The first paragraph of the Discussion section should provide a brief summary of the study and its findings, noting the systematic design of this review.

11- Discussion Conclusions: “ We intend to test out this approach in the near future.” Some specifics on how this will be tested would be useful here.

12- Discussion Conclusions, final sentence: I have no problem with hope, but I think you could be more assertive here.

Comments from Reviewers:

Reviewer #5: I think we still have a difference of opinion about some of the issues. The authors seem to think that as a statistician I am only concerned about bias in the statistical sense but the issues I raised are, as far as I can see, more related to the everyday meaning of the word. So my concern about relying on previous reviews is that if the previous reviewers had a particular unconscious interpretation of the terms they were using then this would affect their presentation. If the authors are convinced that everything is genuinely obvious then there is no issue so I would not wish to press the point.

Michael Dewey

[LINK]

---

## [Editor Report · Decision Letter 4]

11 Sep 2020

Dear Dr. McGill, 

On behalf of my colleagues and the academic editor, Dr. Margaret Kruk, I am delighted to inform you that your manuscript entitled "Qualitative process evaluation from a complex systems perspective: a systematic review and framework for public health evaluators" (PMEDICINE-D-20-00099R4) has been accepted for publication in PLOS Medicine. 

PRODUCTION PROCESS

PRESS

PROFILE INFORMATION

Thank you again for submitting the manuscript to PLOS Medicine. We look forward to publishing it. 

Best wishes, 

Thomas McBride, PhD

Senior Editor 

PLOS Medicine

plosmedicine.org